# The Association between School Climate and Aggression: A Moderated Mediation Model

**DOI:** 10.3390/ijerph18168709

**Published:** 2021-08-18

**Authors:** Zhenhua Li, Chengfu Yu, Yangang Nie

**Affiliations:** Research Center of Adolescent Psychology and Behavior, School of Education, Guangzhou University, Guangzhou 510006, China; lzhpsy@foxmail.com (Z.L.); yuchengfu@gzhu.edu.cn (C.Y.)

**Keywords:** school climate, aggression, self-control, parent-child relationship

## Abstract

While previous studies have shown evidence of an association between school climate and aggression, few have explored the mechanisms behind this association. As such, this cross-sectional study focused on both the mediating effects of self-control and the moderating effects of the parent-child relationship on the association between school climate and aggression. Data were obtained through an anonymous survey conducted among 1030 Chinese elementary and middle school students (52.72% male, average age = 11.53 years), who responded to items on school climate, aggression, self-control, and the parent-child relationship. First, the results showed that school climate was negatively associated with aggression. Second, a mediation analysis showed that self-control significantly mediated the association between school climate and aggression. Third, a moderated mediation analysis showed that the parent-child relationship significantly moderated the first stage of the indirect path (school climate → self-control). Specifically, this association was notably stronger among children and adolescents with better parent-child relationships. In sum, these findings constitute a valuable reference for both improving self-control and in the context of targeted interventions aimed at preventing aggression in children and adolescents in China.

## 1. Introduction

Aggression has remained a hot topic in the field of child and adolescent development. Notably, a survey of 202,056 adolescents across 40 countries found that 10.7% had engaged in aggressive behaviors [1]. Furthermore, a study by Wang et al. [2], which surveyed 1719 junior high school students in Henan, China, reported that 17.9% of the students exhibited aggressive behaviors toward their peers in the past 12 months. This is a critical issue, as aggression is empirically associated with several adverse effects on physical health, mental health, personality development, and social communication [3,4]. This makes it particularly important to explore any contributing factors as well as the internal mechanisms. The current literature shows that a variety of factors can influence aggression, including exposure to community violence [5], parental support [6], and peer victimization [7]. However, there is still a lack of evidence related to the school environment.

Children and adolescents are offered a large variety of opportunities to study and interact while at school. In this context, a positive school climate is undoubtedly an important protective factor against aggression [8,9]. More specifically, school climate is defined as the quality and character of school life, thus reflecting relevant norms, goals, values, interpersonal relationships, teaching/learning practices, and organizational structure [10]. The stage-environment fit theory [11] posits that healthy student development requires interpersonal relationships involving mutual trust and care as well as opportunities for self-expression and independent decision-making. As such, schools that do not provide educational environments that facilitate student development fail to match these needs, which may lead to adverse emotional and behavioral outcomes [12,13,14]. For example, a longitudinal study showed an increase in problem behaviors among adolescents as the school climate declined over time [15]. Meanwhile, other previous studies have shown that school climate is closely related to aggression [8,16]. In this regard, Wang et al. [17] demonstrated that a positive school climate helped reduce aggression. However, there is a significant gap in the scholarly literature concerning which factors mediate the association between school climate and aggression; there is also a lack of information about the various moderators that exacerbate and attenuate this association in the Chinese context.

### 1.1. Self-Control as a Mediator

Self-control is one of the most powerful and beneficial capacities of the human psyche. It is defined as the ability to override or change one’s inner reactions, including the interruption and avoidance of acting on undesired behavioral tendencies [18]. According to the general aggression model (GAM) [19], inputs stemming from situational and personological variables jointly affect the evaluation and decision-making process, thus generating aggressive behaviors. This model is consistent with ecological systems theory, which emphasizes the impacts of external and internal factors on individual-level development [20]. This study was therefore conducted assuming that the school environment (e.g., school climate) may influence aggression through individual factors (e.g., self-control).

Children and adolescents spend most of their waking time at school [21,22]. Based on data from the National Longitudinal Survey of Youth, Turner et al. [23] asserted that the school environment was a critical factor in the development of self-control, specifically demonstrating that the effects of school socialization on self-control were a significant net of parental socialization. Furthermore, Gottfredson and Hirschi [24] suggested that the school environment imposes three main evidential effects on self-control. First, members of the school faculty (especially teachers) are capable of simultaneously supervising multiple students. Second, teachers are committed to maintaining a healthy educational environment, and therefore recognize that antisocial behaviors are harmful for children. Third, many schools and teachers hold the power to maintain order and enforce effective discipline, which theoretically entails authority and agency in punishing the lack of self-control. Previous studies have also shown that different dimensions of the school climate (e.g., cohesion, friction, and overall satisfaction) are significantly related to effortful control [22]. Zhang et al. [25] similarly found a significant positive correlation between school climate and self-control (e.g., effort control). A longitudinal study also demonstrated that self-control was significantly impacted by school classroom characteristics [26]. In sum, substantial evidence suggests that the school climate is an important factor in the development of self-control.

In addition, a large number of studies have shown that self-control is closely connected to aggression [27,28,29]. The general theory of crime [24] entails that the fundamental causes of all crimes and problem behaviors are rooted in the lack of self-control. Compared to individuals with high self-control, those with low self-control are more inclined to exhibit a series of problem behaviors, including aggression, the perpetration of bullying, and violence [28,30,31]. Notably, a previous meta-analysis among Chinese students found a substantial negative link between self-control and aggression, with moderating factors including age, gender, and specific measures of aggression [32]. In this context, provocation is associated with decreased self-control, thereby increasing aggressive behavior. However, this adverse effect can be reduced by increasing the amount of self-control resources [33]. In addition, Denson et al. [27] demonstrated that failures in self-control were associated with aggressive behaviors. Inversely, the ability to induce self-control is strengthened when aggression is reduced. An investigation into the relationships between low self-control, peer delinquency, and aggression among adolescents indicated that both low self-control and peer delinquency could positively predict aggression, while low self-control had an indirect effect on aggression through peer delinquency [34]. As such, the current body of evidence suggests that self-control is a significant protective factor against aggression.

Considering that school climate is related to self-control, which can in turn predict aggression, we speculate that self-control may be a potential mediating mechanism in the association between school climate and aggression. According to Wang et al. [17], deviant peer affiliation plays a mediating role in the relationship between school climate and aggressive behavior. Xu et al. [35] similarly demonstrated that self-control significantly mediated the association between school climate and depression. Moreover, a longitudinal study on the relationship between religion and aggressive behavior found that self-control and empathy fully mediated the association between religion and aggression [36].

Based on the scholarly evidence, we posited the following:

**Hypothesis** **1** **(H1).**
*Self-control significantly mediates the association between school climate and aggression.*


### 1.2. The Parent-Child Relationship as a Moderator

Ecological systems theory entails that family environment is the basic unit in the ecological model of human development. Of particular note, the parent-child relationship is one of the most important variables for child and adolescent development within the family system. There are also interactions between different environmental influences at the individual level [20,37]. In this regard, previous studies have found specific interactions between the school and family environments, which are two of the most direct systems involved in individual development [38,39]. The protective-protective model posits an interaction between two separate protective factors in predicting development outcomes. In regard to outcome variables, the effect of one protective factor may thus vary based on the level of another protective factor. Furthermore, the promotion hypothesis suggests that one protective factor may enhance the association between other protective factors and outcome variables; on the other hand, the exclusion hypothesis suggests that one protective factor may weaken the association between other protective factors and outcome variables [40,41]. In this study, we speculated that the direct and indirect associations between school climate and aggression may be moderated by the parent-child relationship, with higher-level relationships entailing stronger associations.

Previous studies have demonstrated a close connection between the parent-child relationship and self-control. Specifically, research has shown that high-quality parent-child relationships can predict high levels of self-control [42,43]. According to Gottfredson and Hirschi [24], the effects of school socialization on self-control also vary based on the parenting style; in their study, these effects were positive and significant under good parenting styles, but non-significant under poor parenting styles. However, Turner et al. [23] suggested an opposing viewpoint; although these researchers also believed that school socialization had varied effects on self-control due to different parenting environments, their results showed significant effects only under conditions of poor parental rearing. Nonetheless, scholars generally agree that the relationship between the school environment and self-control is affected by the family environment.

Moreover, many studies have demonstrated that the parent-child relationship is a significant protective factor against aggression, delinquency, and other problem behaviors [44,45,46]. On one hand, the parent-child relationship may moderate the association between self-control and aggression. Letcher et al. [47] found that parenting styles moderated the effects of personality characteristics (e.g., temperament) on problem behaviors among early adolescents. A longitudinal study by Kochanska and Kim [48] indicated that the parent-child relationship moderated the relationship between effortful control and internalizing behavior in young children. In a study among preschoolers, Rubin et al. [49] demonstrated that emotional-behavioral undercontrol at age two was significantly and positively correlated with externalizing problems at age four; this association was particularly strong among children with high levels of negative parenting. On the other hand, parent-child relationships may play a moderating role in the association between school climate and aggression. Wallace and May [50] suggested that adolescents with good parent-child relationships were less likely to exhibit strong negative emotions when confronted with bad school environments when compared to those with poor parent-child relationships, which may reduce the occurrence of problem behaviors. Heatly and Votruba-Drzal [39] demonstrated that low engagement upon school entry may eventually result in dysfunctional school behaviors, but that parent-child relationships buffer the negative effects of teacher conflict on engagement. 

According to this literature, we posited the following: 

**Hypothesis** **2** **(H2).***The parent-child relationship significantly moderates the first stage of the indirect path (school climate* → *self-control) and the second stage* of the indirect path *(self-control* → *aggression); these associations are stronger for children and adolescents with higher-quality parent-child relationships.*

**Hypothesis** **3** **(H3).**
*Parent-child relationships significantly moderate the association between school climate and aggression; this association is stronger for children and adolescents with higher-quality parent-child relationships.*


### 1.3. The Goal of This Study

While previous studies have shown that school climate is closely related to aggression, the underlying mechanism is still unclear. At this time, few studies have examined the influences of school and family factors on child and adolescent development, especially in the context of China. In this study, we constructed a moderated mediation model (Figure 1) with the aim of providing useful insights for the prevention and reduction of aggression. Following ecological systems theory, we examined whether self-control mediated the association between school climate and aggression in addition to whether the parent-child relationship moderated the direct and indirect associations between school climate and aggression. 

## 2. Materials and Methods

### 2.1. Participants

We employed random cluster sampling to recruit participants from five elementary and junior middle schools in Guangzhou. A total of 1070 questionnaires were sent out. None of the subjects refused to participate in the test. Subsequently, 40 entries with missing data were eliminated, with an effective rate of 96.26%. The final sample consisted of 1030 children and adolescents (52.72% males) ranging from 8–15 years of age (mean age = 11.53, SD = 1.57). Proportions of 16.60% were from grade 4, 19.71% from grade 5, 19.13% from grade 6, 21.46% from grade 7, 12.82% from grade 8, and 10.29% from grade 9. 

### 2.2. Measures

#### 2.2.1. School Climate

We used the Delaware School Climate Surveys-Student (DSCS-S) to measure school climate [51]. The scale contains a total of 29 items across seven subscales, including teacher-student relations, student-student relations, student engagement school-wide, clarity of expectations, fairness of rules, school safety, and bullying school-wide. Participants were thus asked to evaluate their respective schools based on prompts (e.g., “The school rules are fair”; “Teachers care about their students”). All items were rated on a 4-point scale ranging from 1 (strongly disagree) to 4 (strongly agree), with item scores then averaged to create a composite of the school climate. Here, higher scores indicated better school climates. The result of confirmatory factor analysis (CFA) indicates the scale has good structure validity in this study: *χ*^2^/*df* = 3.33, CFI = 0.88, TLI = 0.87, RMSEA = 0.091, and SRMR = 0.061. Moreover, in this study, the scale received a Cronbach’s α of 0.93. 

#### 2.2.2. Self-Control

We used the Brief Self-Control Scale (BSCS) compiled by Tangney et al. [18] to measure the level of trait self-control. The scale contains a total of 13 items. Participants were asked to indicate how each item pertained to themselves (e.g., “I am good at resisting temptation”; “I have a hard time breaking bad habits”) using a 5-point scale ranging from 1 (not at all) to 5 (very much), with item responses then averaged. Higher scores indicated higher levels of self-control. The result of CFA indicates the scale has good structure validity in this study: *χ*^2^/*df* = 3.24, CFI = 0.92, TLI = 0.89, RMSEA = 0.094, and SRMR = 0.051. Moreover, in this study, the scale received a Cronbach’s α of 0.82.

#### 2.2.3. Aggression

We used the Mini Direct Indirect Aggression Inventory (Mini-DIA) developed by Österman [52] to assess aggression. The scale contains three items (e.g., “Have you kicked, hit, or push someone?”; “Have you gossiped about someone, spread rumors about someone, or socially excluded someone?”). Participants rated each item using a 5-point scale ranging from 0 (never) to 4 (very often), with item scores then averaged. Here, higher scores indicated higher levels of aggression. The result of CFA indicates the scale has good structure validity in this study: *χ*^2^/*df* = 1.14, CFI = 0.99, TLI = 0.99, RMSEA = 0.012, and SRMR = 0.022. Moreover, in this study, the scale received a Cronbach’s α of 0.84. 

#### 2.2.4. Parent-Child Relationship

We used the Closeness to Parents Scale developed by Buchanan, Maccoby, and Dornbush [53] to measure the parent-child relationship. The scale contains nine items. Participants were asked to rate each item based on their actual relationships with their parents (e.g., “How open are you when talking to your parents?”; “How interested are your parents in talking to you when you want to talk?”). All items were rated on a 5-point scale ranging from 1 (not at all) to 5 (very), with item scores then averaged to create a composite score. In this case, higher scores indicate better parent-child relationships. The result of CFA indicates father-child relationship subscale has good structure validity in this study: *χ*^2^/*df* = 4.42, CFI = 0.97, TLI = 0.96, RMSEA = 0.058, and SRMR = 0.027. In addition, the result of CFA indicates mother-child relationship subscale has good structure validity in this study: *χ*^2^/*df* = 3.99, CFI = 0.97, TLI = 0.96, RMSEA = 0.054, and SRMR = 0.027. Moreover, in this study, the total scale received a Cronbach’s α of 0.90. When broken down to focus on the father-child and mother-child relationship subscales, we returned respective Cronbach’s α values of 0.85 and 0.83.

### 2.3. Procedures and Statistical Analyses

All survey materials and procedures were approved by the respective ethics review committees at the universities affiliated with each researcher (protocol code: GZHU2019007). Oral consent was obtained from all participants, including the principals, teachers, parents, and students. The survey was conducted with the class as a study unit. Prior to the assessments, trained researchers explained relevant instructions and precautions to all participants. They also explained the study purpose and emphasized that the survey was anonymous. Participants were asked to complete their self-report questionnaires independently. In case of discomfort, all participants were given the option to withdraw at any time. The questionnaires were collected on-site after completion. The measurement process lasted approximately 30 min. 

We used the IBM SPSS software (version 21.0) for the descriptive statistical analysis. Moreover, mediation and moderation effects were tested with Mplus 7.2 [54]. Missing values were handled via full information maximum likelihood estimation, and bootstrapping analysis with 5000 replicates was performed to verify the significance of the paths. According to Hoyle’s suggestion [55], a model fit is considered acceptable when *χ*^2^/*df* is small than 5, CFI and TLI are larger than 0.90, and RMSEA and SRMR are small than 0.08.

## 3. Results

### 3.1. Descriptive Statistics and Correlations

Table 1 provides the means, standard deviations, and Pearson correlations between the research variables. As shown, school climate was positively associated with self-control and the parent-child relationship, but negatively associated with aggression. Meanwhile, self-control was positively associated with the parent-child relationship, but negatively associated with aggression. Finally, aggression was negatively associated with the parent-child relationship.

### 3.2. Testing for Mediation Effects of Self-Control

The mediation model is represented in Figure 2, which had an excellent fit to the data: *χ*^2^/*df* = 1.53, CFI = 0.996, TLI = 0.983, RMSEA = 0.023, and SRMR = 0.011. After controlling for gender and age, it was found that school climate positively predicted self-control (*β* = 0.36, *SE* = 0.10, *t* = 3.83, *p* < 0.001, 95% confidence interval [CI] [0.23, 0.64]), and self-control negatively predicted aggression (*β* = −0.16, *SE* = 0.04, *t* = −3.78, *p* < 0.001, 95% CI [−0.24, −0.07]). Moreover, the residual effect of school climate on aggression was also significant (*β* = −0.26, *SE* = 0.04, *t* = −5.85, *p* < 0.001, 95% CI [−0.35, −0.17]). Bootstrapping analyses indicated that self-control significantly mediated the relationship between school climate and adolescent aggression (indirect effect = −0.06, *SE* = 0.02, *t* = −2.61, *p* < 0.01, 95% CI [−0.10, −0.02]).

### 3.3. Testing for Moderated Mediation Effects

The moderated mediation model is represented in Figure 3, which had a very good fit to the data: *χ*^2^/*df* = 1.56, CFI = 0.996, TLI = 0.968, RMSEA = 0.023, and SRMR = 0.007. The bias-corrected percentile bootstrap results indicated that the indirect effect of school climate on adolescent aggression through self-control was moderated by parent-child relationship. Specifically, school climate (*β* = 0.27, *SE* = 0.03, *t* = 8.28, *p* < 0.001, 95% CI [0.18, 0.32]) and parent-child relationship (*β* = 0.26, *SE* = 0.03, *t* = 7.92, *p* < 0.001, 95% CI [0.20, 0.34]) showed significant positive association with self-control. Moreover, parent-child relationship moderated the association between school climate and self-control (*β* = 0.08, *SE* = 0.04, *t* = 2.29, *p* < 0.05, 95% CI [0.01, 0.15]). We conducted a simple slopes test, and, as depicted in Figure 4, the positive association between school climate and self-control was significantly stronger for children and adolescents with higher parent-child relationships (1 *SD* above the mean; *β* = 0.34, *SE* = 0.04, *t* = 7.66, *p* < 0.001, 95% CI [0.25, 0.43]) than for those with lower parent-child relationships (1 *SD* below the mean; *β* = 0.21, *SE* = 0.04, *t* = 5.69, *p* < 0.001, 95% CI [0.14, 0.28]). Furthermore, self-control had a significant negative association with aggression (*β* = −0.16, *SE* = 0.04, *t* = −3.54, *p* < 0.001, 95% CI [−0.23, −0.06]). However, the interaction between school climate and parent-child relationship (*β* = 0.01, *SE* = 0.04, *t* = 0.21, *p* = 0.833, 95% CI [−0.23, −0.06]), and the interaction between self-control and parent-child relationship (*β* = −0.06, *SE* = 0.04, *t* = −1.44, *p* = 0.149, 95% CI [−0.15, 0.02]) in predicting aggression were not significant.

Finally, the bias-corrected percentile bootstrap results indicated that the indirect link between school climate and aggression via self-control was stronger for children and adolescents with high parent-child relationships (indirect effect = −0.08, *SE* = 0.03, 95% CI [−0.15, −0.03]) than for those with low parent-child relationships (indirect effect = −0.02, *SE* = 0.01, 95% CI [−0.05, −0.002]). Therefore, the mediating effect of self-control between school climate and adolescent aggression was moderated by parent-child relationship.

## 4. Discussion

This study employed a moderated mediation model to assess the psychological mechanisms behind the association between school climate and aggression in the Chinese context. Our findings constitute valuable references for interventions and other approaches aimed at preventing aggression. We have divided these into three discrete components, with limitations and implications following.

### 4.1. The Relationship between School Climate and Aggression

We found a negative relationship between school climate and aggression, which is consistent with previous research [8,17]. According to the stage-environment fit theory, behavioral development is influenced by the school environment. If the school fails to provide an educational environment that is suitable for individual development, then this is likely to lead to problem behaviors among students (e.g., aggression). To the contrary, a positive school climate can provide students with safety, instill mutually supportive interpersonal relationships, and offer opportunities for self-expression. In this atmosphere, children and adolescents tend to develop positive school attitudes and emotions (e.g., a strong sense of belonging), thus facilitating them in social adjustment [11,56]. As a result, students are less likely to engage in aggressive behaviors [16,57]. In sum, it is clear that positive school climates are important protective factors against aggression in children and adolescents.

### 4.2. The Mediating Effect of Self-Control

Our results confirmed that self-control significantly mediated the relationship between school climate and aggression. On the one hand, we found that the school climate was positively associated with self-control, which supports previous research showing that positive school climates are important environmental factors in the proper development of self-control [25]. According to Gottfredson and Hirschi [24], schools and teachers have the authority and means to punish the lack of self-control. Furthermore, teachers are committed to maintaining a healthy educational environment. Given these conditions, safe and orderly schools that promote harmonious teacher-student relationships can provide necessary support while helping to improve self-control among students. Schools with good climates also tend to pay more attention to educational provisions concerning the self-control ability, and teach related courses on self-control and social skills [58], which are conducive to the development of self-control.

On the other hand, self-control was negatively associated with aggression. Children and adolescents with higher self-control reported less aggressive behavior, which is consistent with previous research [28,33]. According to the general theory of crime, the lack of self-control is a vital reason for crimes and other problem behaviors. In this regard, individuals with low self-control are more likely to exhibit aggressive behavior due to difficulty restraining impulsive emotions and behavioral reactions. In addition, the self-control resource model posits that self-control resources are limited. When ego depletion occurs during task completion or goal achievement, it becomes difficult to implement effective self-control, which can lead to various undesirable behavioral results, including increased aggression [59] and violent behaviors [31]. As such, self-control is the core protective factor against aggression. School climate has an indirect effect on aggression through self-control. Our study further uncovered the potential mediating mechanism in the association between school climate and aggression.

### 4.3. The Moderating Effect of the Parent-Child Relationship

Our results confirmed that the parent-child relationship significantly moderated the first stage of the indirect path (school climate → self-control). Notably, the association between school climate and self-control was stronger for children and adolescents with high-quality parent-child relationships than for those with low quality parent-child relationships. In the first stage of the mediation path, high-quality parent-child relationships promote the association between school climate and self-control, which is consistent with the views of Gottfredson and Hirschi [24]. Previous studies have demonstrated that positive parent-child relationships play significant roles in self-control [42,43]. According to attachment theory, continued successful communication and safe bonding between parents and children facilitate children in developing the ability to regulate their emotions, which is an important component of self-control [60]. According to ecological systems theory, the school and family systems jointly influence child and adolescent development. The protective factor-protective factor model also suggests the existence of an interaction between two different protective factors in predicting development outcomes. In this study, both the school climate and parent-child relationship were protective factors for self-control. Moreover, the school climate was positively correlated with self-control. Notably, high-quality parent-child relationship enhanced this positive association, which supports the promotion hypothesis of the protective factor-protective factor model.

Surprisingly, the parent-child relationship did not moderate the relationship between self-control and aggression. First, according to the general theory of crime [24], the fundamental causes of crimes and problem behaviors (e.g., aggression) are rooted in the lack of self-control. Although parent-child relationships are also associated with internalization issues in adolescents (e.g., depression) [61] and externalized problem behaviors (e.g., aggression and antisocial behavior) [44,62], the effect of self-control on aggression may be more significant than the effect of parent-child relationship. Second, the association between self-control and aggression might be moderated by other variables (e.g., parent-child communication and parental warmth). Furthermore, the parent-child relationship failed to moderate the relationship between school climate and aggression. This may be because Chinese students spent most of their time at school. As aggressive behaviors occur more frequently in places outside the family environment (e.g., schools), the effects of school climate on aggression may be greater than those of the parent-child relationship. As such, the parent-child relationship does not sufficiently influence the dominant role of the school climate in aggression.

### 4.4. Limitations and Future Research

This study also had some limitations. First, although we discussed correlations between variables, we solely employed a cross-sectional design, and could therefore not strictly confirm any causal relationships. As such, future research should implement a longitudinal design involving multiple time points to further investigate the causal relationships. Second, our data were obtained via self-reported questionnaires, which may have entailed common method bias. In future studies, researchers should collect data through multiple reporters (e.g., teachers, parents, and peers) and assessment methods (e.g., observations and interviews). Third, various components of the school climate (e.g., teacher-student relationships and school safety) may have different influences on aggression. Future research should explore the relationships between different dimensions of the school climate and aggression, thereby revealing their underlying mechanisms. Fourth, aggression was measured using a three-item scale, which may not be sufficient to examine aggression fully. In the future, widespread scales with more items can be used, such as the Buss-Warren Aggression Questionnaire (BWAQ) [63,64]. Finally, all study participants were from South China. As China is a multi-ethnic and multicultural country, the school environment, family environment, and personality factors may vary based on the region. In this case, the universality and applicability of this model should be investigated in other regions of China.

### 4.5. Implications

This study jointly employed school and family factors via a moderated mediation model to investigate the mechanism behind aggression, which has important implications for both prevention and intervention. First, we confirmed that the school climate is a core protective factor against aggression. This emphasizes that schools should strive to create safe, fair, and harmonious environments for students, promote communication between teachers and students, and effectively guide and educate students who engage in aggressive behaviors. Second, we found that self-control is the mediation mechanism in the association between school climate and aggression. Schools and teachers should therefore focus on the cultivation of self-control, incorporate self-control-related courses into the teaching system, and exercise the self-control ability during practical activities (e.g., aerobic exercise and mindfulness practices) [65], which may help reduce aggression. Third, educators should be aware of the importance of school climate (e.g., teacher-student relationship, peer relationship) and self-control in the healthy development of children and adolescents. During routine classroom teaching activities, teachers can adopt themed class meetings, academic lectures, games, and other activities to cultivate students’ interpersonal skills and self-control. In addition, schools can create a good school climate, improve students’ self-control, and emphasize the harms of aggressive behaviors through brochures, psychological activity months, interpersonal communication, and group counseling social media (e.g., WeChat official accounts). Finally, we found that the parent-child relationship can promote the association between the school climate and self-control. Thus, parents should frequently communicate with their children while promoting mutual trust and respect. Children must also be provided with a sense of security, which entails the cultivation of positive and healthy parent-child relationships. At the same time, schools and parents should strengthen their cooperation and communication, and provide support and encouragement to children and adolescents. In sum, they should jointly work to create a good atmosphere that promotes healthy child and adolescent development.

## 5. Conclusions

In this study, we found that the school climate was positively associated with aggression, while self-control significantly mediated the relationship between school climate and aggression. Moreover, the parent-child relationship significantly moderated the first stage of the indirect path. Of particular note, this association was much stronger for children and adolescents with higher-quality parent-child relationships than for those with lower-quality parent-child relationships. Taken together, these findings strengthen the current scholarly understanding of the potential mediating and moderating mechanisms of the school climate on child and adolescent aggression.

## Figures and Tables

**Figure 1 ijerph-18-08709-f001:**
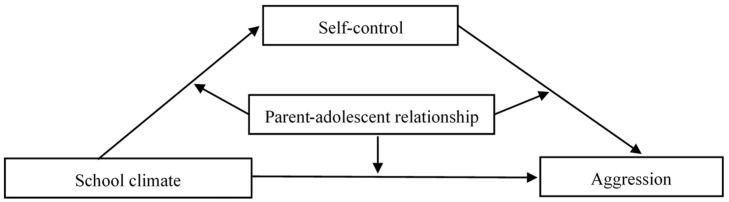
The proposed moderated mediation model.

**Figure 2 ijerph-18-08709-f002:**
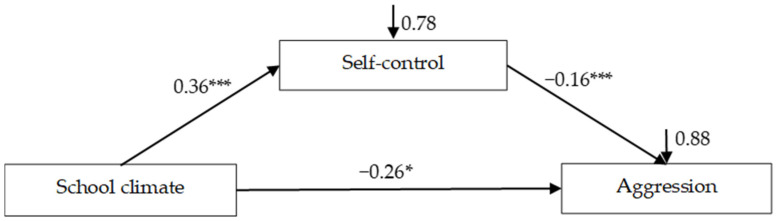
Model of the mediating role of self-control between school climate and aggression. Values are standardized coefficients. * *p* < 0.05, *** *p* < 0.001.

**Figure 3 ijerph-18-08709-f003:**
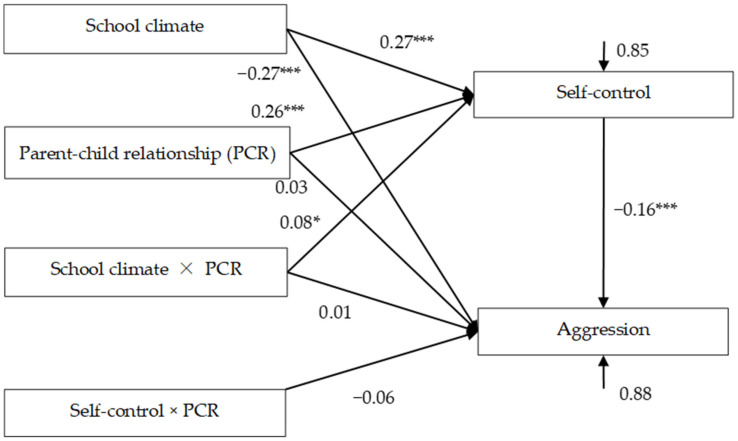
Model of the moderating role of parent-child relationship on the indirect relationship between school climate and aggression via self-control. Values are standardized coefficients. * *p* < 0.05, *** *p* < 0.001.

**Figure 4 ijerph-18-08709-f004:**
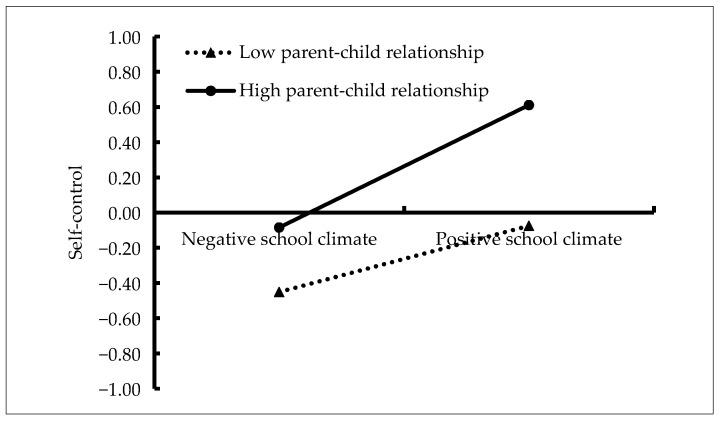
Self-control as a function of school climate and parent-child relationship.

**Table 1 ijerph-18-08709-t001:** Descriptive statistics and correlations among variables.

Variables	*M*	*SD*	1	2	3	4
1. School climate	3.27	0.44	1			
2. Self-control	3.65	0.69	0.34 ***	1		
3. Aggression	0.62	0.75	−0.31 ***	−0.24 ***	1	
4. Parent-child relationship	3.76	0.71	0.35 ***	0.30 ***	−0.11 ***	1

Note: *** *p* < 0.001.

## Data Availability

The data presented in this study are available on request from the corresponding author.

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
