# Peer review of "The Association between School Climate and Aggression: A Moderated Mediation Model"

_ijerph, 2021, doi:10.3390/ijerph18168709_

Round 1

Reviewer 1 Report

Minor revision of the bibliography:

- R546, reference 31......China instead of.... china.

Author Response

Response: Thank you for your valuable suggestion. We have modified the bibliography.

Reviewer 2 Report

This is a very well-written report on the association between school climate and aggression. Also authors, examine the mediating effects of self-control and the moderating effects of the parent-child relationship on the association between both school climate and aggression. I am very impress about the quality of this manuscrip. The analyses conducted are clearly explained and the contribution to research is properly elaborated. The paper make a significant contribution to the research which have the potential to inform practice in important ways. Minor recommendations are suggested in order to enhance the usefulness of this manuscript.

Introduction: Readers will benefit on how rates of aggression in China differ from other region in the world.

Secondc, In the 2.3 section authors suggest that “All survey materials and procedures were approved by the respective ethics review committees at the universities affiliated with each researcher”. It would be recommended if authors could include the number/code of committees, and if available if registered.

Also, I suggest to provide more details about the process of obtaining consent to participate in this study within the method section. Please, inculde the number of participants who were approached, the number of patients who refused to participate as well as missing data. Also, how were participants selected? There were some inclusión/exlusion criteria?

Last, How educators should take into account the present findings in the routine of classroom? And how results fit with future intervention approach within school settings?

Despite this limitations, the core of the article is important and relevant and I suggest the paper has a huge potential for publication.

Reviewer 3 Report

The manuscript "The Association Between School Climate and Aggression: A Moderated Mediation Model" addresses a relevant and timely issue, i.e. risk and protective pathways towards peer aggression considering both personological and environmental variables. This study has several merits, including a very good sample size and a generally good consistency between the theoretical background, research questions and the statistical methods adopted to investigate them. However, I have some -mainly methodological - concerns which I think should be addressed before the study can be accepted for publication.

First, considering that the main focus of the study was on correlates of school climate, it is not clear why this construct was considered as monodimensional, especially as the scale adopted to assess school climate was multidimensional including seven subscales. This is recognized as a limitation in the manuscript, but this choice should also be justified. Was reliability problematic for singular subscales? Has this scale already been used as mono-dimensional? It seems like it includes very different subscales (take, e.g., student-student relations, school safety, bullying school wide). This also leads to confusing expressions, like "school climate positively predicted aggression", as "school climate" by itself does not mean much.

Although I appreciate the mediation model and its strict relationship with the hypothesis and theoretical background, it is not clear what kind of model was used. Referencing the SPSS package is not sufficient, Authors here should specify what models were used and why. Was structural equation modeling used? The estimation method (e.g. WLSMV or MLR) should be specified together with the rationale for this choice. Fit measures (see e.g. http://www.davidakenny.net/cm/fit.htm) should also be reported. In line with this, I also think that a CFA would be preferable to presenting Cronbach's alphas, which are not very informative especially considering the high numerosity of the sample (see, e.g., https://doi.org/10.1037/met0000144). This would also allow to present a covariance matrix instead of the correlations that were presented (by the way, are they Pearson correlations? Were distributional assumptions checked?)

I think the fact that the outcome variable of the model (aggression) is measured by a three-items scale should be acknowledged as a serious limitation of this study.

Besides these more serious concerns, I have some minor observation that you find listed below:

Lines 14-15 read: "First, the results showed that school climate positively predicted aggression". Besides the issue concerning the choice of considering school climate as a monodimensional variable, this formulation is not clear, and seems to contradict the results section ("school climate was positively associated with self-control and the parent-child relationship, but negatively associated with aggression", lines 252-253).

Lines 16-19 read: "Third, a moderated mediation analysis showed that the parent-child relationship significantly moderated the following mediation path: school climate → self-control → aggression. Fourth, this mediating effect was notably stronger among children and adolescents with better parent–child relationships". If I understood correctly, the fourth point is actually an explanation of point three. If I'm correct in this interpretation, it should not be presented as a new result.

The phrase "the essence of the school climate" (line 75) is confusing to me. I would say something like "different dimensions of school climate".

"Further" is used as a synonym of "furthermore" or "moreover" in several sentences throughout the manuscript. This should be corrected.

In the Participants section, the sample is presented as composed of early adolescents, but the age ranges from 8 to 15 years. I would say "children and adolescents" instead.

It is not clear what time window was specified for the Mini-DIA: did it refer to, e.g., the last 3 months?

Finally, the mediation of self-control was emphasized as one of the most important results of the study, but the indirect effect presented are very small (< 0.1), and account for a small portion of the total effect of school climate on aggression. I think, regardless of statistical significance, the effect sizes should be considered when discussing the practical significance of the results.

Round 2

Reviewer 3 Report

I have no additional comments.